# A keystone mutualism underpins resilience of a coastal ecosystem to drought

Christine Angelini[1], John N. Griffin[2], Johan van de Koppel[3,4], Leon P.M. Lamers[5], Alfons J.P. Smolders[5], Marlous Derksen-Hooijberg[5], Tjisse van der Heide[5] & Brian R. Silliman[6]

Droughts are increasing in severity and frequency, yet the mechanisms that strengthen ecosystem resilience to this stress remain poorly understood. Here, we test whether positive interactions in the form of a mutualism between mussels and dominant cordgrass in salt marshes enhance ecosystem resistance to and recovery from drought. Surveys spanning 250 km of southeastern US coastline reveal spatially dispersed mussel mounds increased cordgrass survival during severe drought by 5- to 25-times. Surveys and mussel addition experiments indicate this positive effect of mussels on cordgrass was due to mounds enhancing water storage and reducing soil salinity stress. Observations and models then demonstrate that surviving cordgrass patches associated with mussels function as nuclei for vegetative re-growth and, despite covering only 0.1–12% of die-offs, markedly shorten marsh recovery periods. These results indicate that mutualisms, in supporting stress-resistant patches, can play a disproportionately large, keystone role in enhancing ecosystem resilience to climatic extremes.

[1] Department of Environmental Engineering Sciences, Engineering School for Sustainable Infrastructure and Environment, University of Florida, PO Box 116580, Gainesville, Florida 32611, USA. [2] Department of Biosciences, Swansea University, Singleton Park, Swansea SA2 8PP, UK. [3] Department of Estuarine and Delta Systems (EDS), Royal Netherlands Institute for Sea Research and Utrecht University (NIOZ-Yerseke), Postbus 140, Yerseke 4400 AC, The Netherlands. [4] Conservation Ecology, Groningen Institute for Evolutionary Life Sciences, University of Groningen, PO Box 11103, Groningen 9700 CC, The Netherlands. [5] Department of Aquatic Ecology and Environmental Biology, Institute for Water and Wetland Research, Radboud University, Heyendaalsweg 135, Nijmegen 6525 AJ, The Netherlands. [6] Department of Marine Science and Conservation, Nicolas School of the Environment, Duke University, 135 Duke Marine Lab Road, Beaufort, North Carolina 28516, USA. Correspondence and requests for materials should be addressed to C.A. (email: c.angelini@ufl.edu).

D roughts are eliciting alarming declines in food production, air quality, carbon storage and biodiversity worldwide[1,2]. In coastal regions where >40% of the global human population resides, the lack of rainfall during drought reduces freshwater discharge and interacts with persistent evapotranspiration to elevate the salinity of estuarine waters and desiccate otherwise moist wetland soils[3–6]. These stressors often act together with disease and consumer outbreaks to cause widespread mortality of dominant plants and reef-building fauna, resulting in shifts to undesirable ecosystem states defined by declines in habitat structure, biodiversity and ecosystem functioning[4–7]. In light of these changes, identifying which factors enhance ecosystem resilience to drought—their ability to resist change and/or rapidly return to desirable states[8]—has emerged as an urgent goal because many communities depend on these coastal habitats for food, protection from storms, water quality enhancement, tourism and other valuable services[9,10]. Ecological theory and perspectives on conservation hypothesize that positive interactions, such as mutualism and facilitation, should promote ecosystem resistance to and recovery from intensifying climatic stress[11–13]. However, evidence that such interactions play an important role in promoting ecosystem resilience to large-scale disturbances is scarce.

Here, we investigate the potential for a mutualism between cordgrass, *Spartina alterniflora*, and the ribbed mussel, *Geukensia demissa*[14,15], to increase salt marsh resistance to drought and fuel subsequent recovery. In the southeastern US, three severe droughts over the past 17 years have caused pervasive die-off of *Spartina alterniflora* (hereafter referred to as cordgrass), the region's dominant, marsh-structuring plant[5,16]. During drought, typically waterlogged marsh soils can dry and oxidize. Coincident with these changes, porewater salinity, acid and heavy-metal concentrations often increase around cordgrass roots, and snail grazing impacts on cordgrass leaves intensify[4,5,17]. Where these stressors reach critical levels within marsh landscapes, they generate die-offs—largely denuded mudflats—that span 10's to 10,000's of square metres[4,5,16]. As severe drought is predicted to occur more frequently with climate change, an overall decline in salt marsh habitat coverage, quality and, hence, ecosystem service provisioning in this region is likely[10,16]. It is therefore imperative to identify the mechanisms that facilitate rapid cordgrass recovery and bolster marsh resilience to this stress.

Across higher elevation, interior marsh platforms where drought-induced die-offs occur[4,5], *Geukensia demissa* (hereafter referred as mussels) form dense mounds of up to 100 individuals around cordgrass stems[15,18]. Within any 10 × 10 m area, there are typically between 1 and 20 mounds that cover 0.1–12% of the platform surface[15,18]. Prior studies indicate that cordgrass serves as a settlement substrate, reduces temperatures via canopy shading and provides nutritional resources to facilitate mussels, while mussels enhance nitrogen availability and aerate soils to stimulate cordgrass growth[14,18,19], but this facultative mutualism has yet to be investigated under drought conditions. Anecdotal observations during the severe droughts of 1999–2001 and 2006–2007 suggest that distinct patches of cordgrass survive within otherwise denuded die-offs and that these patches often may be associated with mussels. Hence, we hypothesized that the mussel–cordgrass mutualism increases cordgrass resistance to drought and does so by a currently undocumented mechanism of facilitation: alleviation of drought-induced soil stress within mounds.

In addition, observational studies[20,21] indicate that cordgrass seed viability and seedling survivorship are very low within die-offs, implying that recolonization primarily occurs through the lateral, vegetative growth of clones that survive along die-off borders and in internal, remnant patches[20]. We verified this

dependence of recovery on clonal growth and also tested the prediction that the mussel–cordgrass mutualism, in increasing the number and spatial dispersion of remnant cordgrass patches that function as sources for re-growth, increases the rate of marsh transitions from mudflat- to cordgrass-dominated states when drought conditions subside.

Results from surveys, field experiments and models reveal that mussels indeed ameliorate soil stress and enhance cordgrass resistance to drought and, in sustaining remnant patches that laterally expand, accelerate marsh recovery after die-off. Thus, the cordgrass–mussel mutualism strongly enhances salt marsh resilience. These findings support theoretical predictions that positive interactions increase ecosystem resilience to large-scale disturbance and advocate for the inclusion of mutualisms in strategies to maintain healthy, resilient ecosystems in the face of increasingly extreme climatic events.

## Results

**Mutualism effects on ecosystem resistance to drought**. To begin to test the hypothesis that mussels increase cordgrass resistance to drought, we surveyed nine salt marshes experiencing die-off (for example, Fig. 1a) that spanned 250 km of southeastern US coastline at the conclusion of a severe drought in June 2012 (Supplementary Fig. 1, Supplementary Table 1). At each marsh, we measured the spatial extent of all die-offs, as well as the area of each mussel-associated cordgrass patch, cordgrass patch without mussels and denuded mussel mound (that is, where stems associated with mussels had died) remaining in each die-off (Supplementary Fig. 2). Patches consisted of distinct clusters of stems between 0.01 and 1.5 m$^2$ in area that survived within die-offs and were categorized as being 'associated with mussels' if one or more stems were positioned <10 cm from a mussel (Fig. 1b), a distance shown in a field experiment to be a conservative estimate of the lateral extent of the mussel–cordgrass interaction (Supplementary Fig. 3). Although cordgrass patches covered only 1.9 ± 0.4% (mean ± standard error of all sites) of die-offs surveyed, we found that the probability of cordgrass surviving within die-offs was markedly higher associated with mussels (64.3 ± 27.4%) than not associated with mussels (1.0 ± 0.06%) across all sites (linear model, Site (Mussels): $t = -6.99$, $P = 0.0001$, Fig. 1c,d). Hence, this survey suggested that mussels enhanced cordgrass' ability to endure lethal drought-induced conditions throughout this region.

**Mechanisms of facilitation**. We then explored whether mussel mounds locally protect cordgrass from drought by increasing soil water storage and reducing porewater salinity stress. We focused on this mechanism because drying of marsh soils is the critical first step in triggering increased soil water deficits and concomitant stressors that are associated with cordgrass die-off during drought[4,5,16,17]. From replicate cores extracted from natural marsh areas with and without mussels, we found that the volume of water stored per unit soil volume increases from an average of 58 to 72%, where mussels are present (Fig. 2a, $t$-test: $t = 6.5$, $P = 0.002$). In addition, weekly monitoring of paired wells positioned in 10 natural cordgrass plots associated and not associated with mussels from May–August 2012 indicate that mussels buffer changes in salinity and do so relatively more as surrounding soils become drier (Fig. 2b, linear regression: $r^2 = 0.39$, $P = 0.001$).

To test whether mussel mounds indeed lower salinity and validate their role in enhancing cordgrass resistance to drought, we performed a mussel addition field experiment. Results revealed that adding mussel mounds around cordgrass stems reduced porewater salinity by 7.7 and 6.4 p.p.t., on average, at the

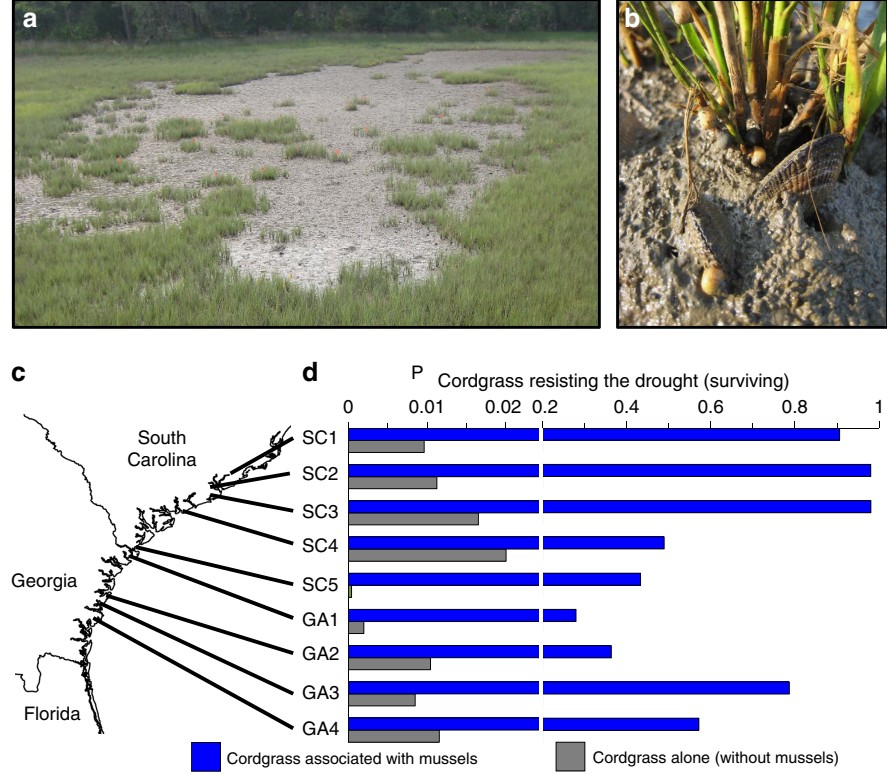

**Figure 1 | Mussels increase cordgrass resistance to drought.** A drought-induced die-off (**a**) and cordgrass growing in association with mussels (**b**). Across nine salt marshes spanning 250 km of coastline (**c**), surveys reveal the probability of cordgrass surviving within drought-induced die-offs (**d**) is markedly enhanced where plants are growing in association with mussels (blue) than growing alone (grey). Please note break in the x-axis in **d**.

surface (0–5 cm) and sub-surface (5–15 cm) cordgrass rooting zones, respectively, relative to control areas without mussels (Fig. 2c, $t$-tests: $t \geq 3.6$, $P \leq 0.05$). In both the experiment and weekly monitoring of paired wells, salinity only reached levels shown to be associated with cordgrass mortality during drought ($> 48$ p.p.t.) (ref. 4) in marsh areas without mussels. We suspect mussels locally enhance water retention through their facilitation of crabs that create belowground water-storage compartments by excavating burrows[18,22], and buffer salinity changes by paving the marsh surface with their shells, which slows evaporative water loss[23]. This field experiment also revealed that mussels stimulate aboveground cordgrass biomass by 190% (Fig. 2d, $t$-test: $t = -8.4$, $P = 0.05$). Altogether with our survey results indicating that mussel mounds substantially enhance cordgrass survival within drought-induced die-offs, these experimental and observational findings provide evidence that mussel amelioration of soil stress, potentially in combination with their enhancement of cordgrass growth before drought occurs, is increasing cordgrass resistance to drought across this region.

**Mutualism effects on ecosystem recovery from drought.** First, to verify observations that clonal expansion, rather than seed dispersal, is the primary reproductive mode by which cordgrass recolonizes die-off mudflats, we used landscape fabric to exclude clonal ramets from penetrating 10 of 20, 1 $m^2$ plots positioned within a die-off and immediately adjacent to bordering cordgrass monocultures. After 1 year, control plots exposed to cordgrass ramets had recovered to $33 \pm 4$ (mean ± s.e.m.) % cover, while those dependent on seeds for cordgrass establishment remained totally bare (Ramet Treatment $t$-test: $t = 6.7$, $P < 0.0001$). Similarly, after 2 and 3 years, an average of $> 100$ live ramets had emerged within ramet control plots, while ramet exclusions had

zero seedlings (Ramet Treatment × Time repeated-measures analysis of variance; Time: $t = -4.09$, $P = 0.0003$; Treatment × Time: $t = 2.67$, $P < 0.0001$, Supplementary Fig. 4).

To then investigate the potential importance of the mussel–cordgrass mutualism in regulating marsh recovery, we first examined if mussels locally affect cordgrass recolonization. With on-the-ground surveys, we monitored the expansion of 79 natural patches that varied in the number of mussels with which they were associated. Although mussel density had a subtle positive effect on natural patch expansion in marsh die-off areas after six (linear regression: $r^2 = 0.14$, $P < 0.0001$) and 12 months (linear regression: $r^2 = 0.12$, $P = 0.0001$), this effect faded after 19 months (Supplementary Table 2, Supplementary Fig. 5). Importantly, however, 77 of the 79 natural patches expanded over time, indicating that drought-resistant patches—including those not associated with mussels—generally function as sources from which cordgrass recolonizes mudflats. Thus, the primary mechanism by which the mussel–cordgrass mutualism appears to mediate marsh recovery is that mussels, in increasing plant resistance to drought-related mortality, markedly enhance the number and spatial dispersion of nuclei from which habitat can regrow within die-off mudflats.

Next, we developed a stochastic cellular automaton model to assess the importance of this mutualism in mediating landscape-scale marsh recovery following die-off. We were particularly interested in exploring how both the timescale of recovery and the contribution of surviving cordgrass patches to recovery shift with two naturally varying die-off features: the size of the die-off and the spatial distribution of surviving patches they contain. In this model, we represent the die-off with a square grid in which each cell corresponds to 0.25 $m^2$ of marsh that is occupied either by mudflat or cordgrass. To track recovery from die-off borders

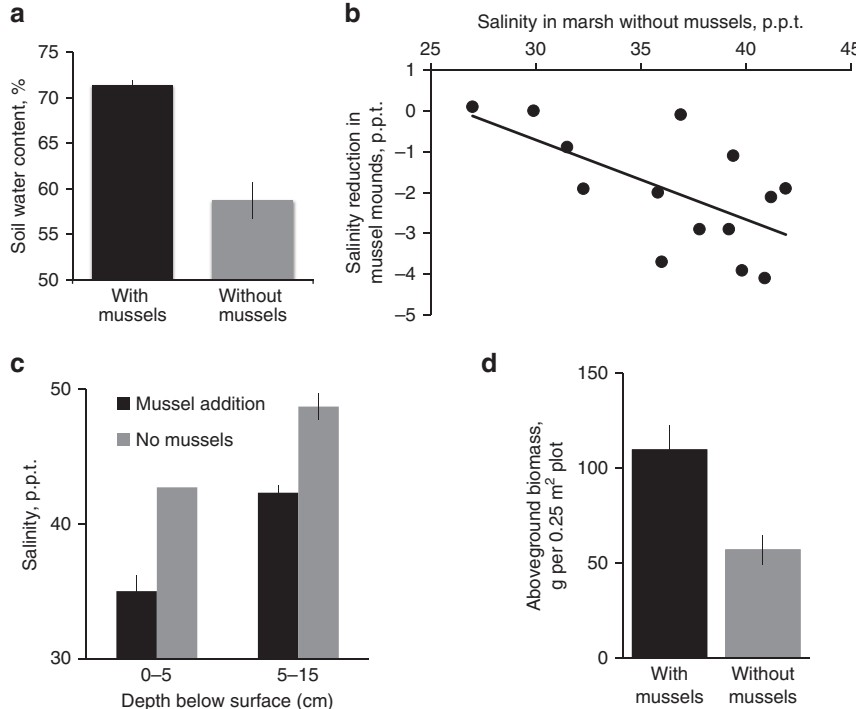

**Figure 2 | Mussel mounds reduce soil salinity stress and enhance cordgrass biomass.** Soil core analyses and weekly monitoring of porewater salinity reveal that naturally occurring mussel mounds within Sapelo Island, GA marsh platforms locally enhance soil water content (**a**) (t-test: $t = 6.5$, $P = 0.002$) and buffer increases in salinity, especially as the salinity in adjacent marsh areas increases (**b**) (linear regression: $r^2 = 0.39$, $P = 0.001$). Experimental mussel addition further shows that mussel mounds reduce salinity in surface and sub-surface cordgrass rooting zones (**c**) (t-test: $t \geq 3.9$ $P \leq 0.05$) and increase aboveground biomass (**d**) (t-test: $t = -8.4$, $P = 0.05$). Data are shown as the mean ± mean standard error for 6 replicate cores in **a** and 3 replicate 0.25 m² plots per treatment in **c** and **d**. There was no variation among replicate salinity measurements collected at 0–5 cm depth in no mussel control plots in **c**. Mussel addition plots are represented in black bars; control, no mussel plots are in grey.

versus patches, we assigned cordgrass cells lining the border of the die-off a cell value of 1, cordgrass patch cells distributed within the die-off a value of 2, and mudflat cells a value of 0. To establish cell transition probabilities, we marked 0.25 m² plots positioned adjacent to cordgrass borders and adjacent to patches in die-off mudflats, and 0.25 m² plots positioned > 2 m away from surviving vegetation in die-off mudflats. After 1 year, we assessed the proportion of plots that changed states and used these values as cell transition parameters in the model (see the 'Methods' section for details). Although cordgrass border expansion can vary with fluctuations in rainfall and patch expansion can shift with mussel (Supplementary Fig. 5, Supplementary Table 2), snail grazer and plant competitor densities[20], we assume that cell transition probabilities remain constant over time because we designed the model to simply explore the role of surviving patches versus bordering vegetation in moderating landscape-scale marsh recovery rather than generate realistic estimates of die-off recovery intervals.

To evaluate the relative importance of remnant cordgrass area versus the configuration of that area in regulating recovery, we distributed cordgrass at the start of each simulation in either: (1) borders only, (2) borders + non-dispersed patches (that is, patch cells aligned along borders), (3) borders + randomly dispersed patches, (4) borders + clustered patches, using a Brownian motion-based fractal random number generator[24] or (5) borders + uniformly dispersed patches. We explored random and clustered patch patterns because Ripley's K analyses[25] of field-collected data indicate patches surviving within die-off areas exhibit these spatial configurations (Supplementary Fig. 6). Similarly, we simulated the recovery of die-offs with uniformly distributed patches to compare cordgrass recolonization of

naturally occurring die-offs (that is, those with only cordgrass bordering, or with cordgrass bordering and in random or clustered patches) to restoration projects that typically plant uniform—that is, regularly spaced—arrays of marsh grass plugs within denuded areas[26]. For each simulation, we derived the number of time steps (years) until the die-off recovered to cordgrass dominance, and calculated the proportion of recovered mudflat area originating from cordgrass patches.

The model predicts that large, 2,000 m² die-offs take 103 and 101 years to recover if they have no patches or have patches that are all aligned to borders, but recover in 22, 13 and 9 years if they have patches dispersed in clustered, random and uniform distributions, respectively (Fig. 3). Furthermore, we found that the proportional contribution of patches to cordgrass recolonization increases substantially with increasing die-off size (Supplementary Fig. 7) because, as die-offs increase in size, the mean distance between each mudflat cell and its closest border cell increases, resulting in bordering vegetation contributing relatively less to recovery. Similarly, the proportional contribution of patches to cordgrass recolonization increases with increasing patch dispersion because as patches become more uniformly dispersed, the mean distance between each mudflat cell and its closest patch cell decreases. Accordingly, with more patch dispersion, there is a higher probability that a mudflat cell will be colonized from an expanding patch rather than from a border. Although this model provides a crude description of patch dynamics, the pronounced, relative differences between model scenarios and evidence from July 2015 surveys indicating that die-offs with many drought-resistant patches are recovering faster after 3 years than those with few (Supplementary Fig. 8) highlight the powerful role these habitat growth nuclei play in regulating

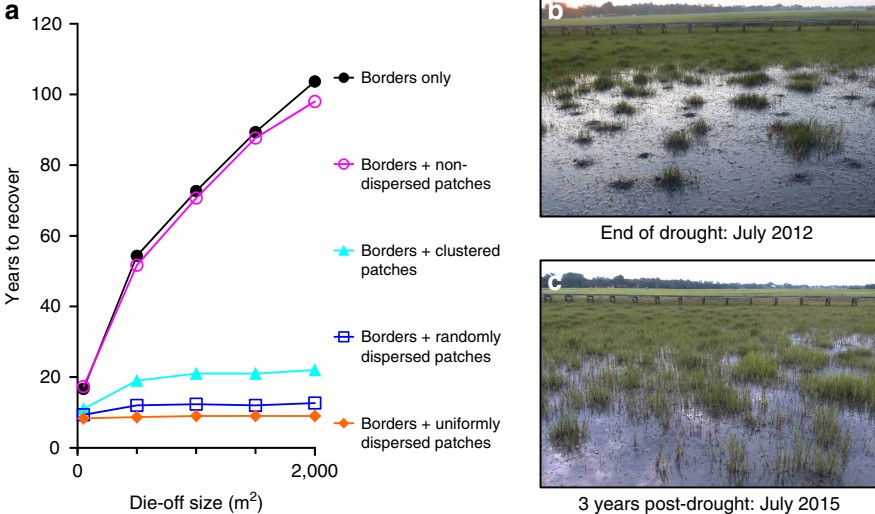

**Figure 3 | Stress-resistant patches increase cordgrass recovery after drought.** Model simulations indicate cordgrass patches accelerate marsh recovery and that this positive effect increases with die-off size and patch dispersion (**a**). A South Carolina marsh with high initial cordgrass patch cover and dispersion at the end of the 2012 severe drought (**b**) and 3 years post-drought (**c**). Data are shown as the mean of three simulation runs per die-off size and patch distribution combination in **a**; different colours and symbols denote the type of patch dispersion as indicated in the legend.

ecosystem transitions from disturbed to healthy states. In following, these findings suggest that efforts to enhance the dispersion and cover of patches within die-off mudflats after climatic stressors subside can markedly accelerate marsh recovery at the landscape scale.

## Discussion

As the above results show, stress-resistant patches can have a disproportionately large effect on ecosystem recovery by virtue of the fundamental spatial processes they support. This finding reveals that: (1) a surprising level of ecosystem resilience can result even when only a limited area of dispersed patches of habitat-forming species remain after episodes of severe stress (that is, a system with low overall resistance), and (2) the time it takes for such ecosystems to recover and, hence, their resilience[8] may not be predicted simply by the size of the disturbance. Instead, to develop accurate predictions for the many systems where dominant, habitat-forming species exhibit vegetative growth and/or spatially limited propagule dispersal (for example, reef-building corals, oysters and mussels; meadow-forming prairie, marsh and sea-grasses; bed-forming marine and riverine macroalgae), ecologists, ecosystem managers and conservation scientists should detail the distribution of remnant patches acting as habitat growth nuclei and identify the mechanisms underpinning such patterns in survival. In salt marshes, our study reveals that these patterns in survival can be largely dictated by a keystone mutualism between cordgrass and mussels, classified as keystone because mussel impacts on ecosystem resistance and recovery are disproportionate to their low cover in marsh landscapes[27,28]. Importantly, recent work showing that mussels do not protect cordgrass against runaway consumption by herbivorous crabs in New England salt marshes[29] serves as a warning that while keystone mutualists may enhance resilience to certain stressors, they may not do so for others, or may have little effect in cases where stress levels are so high that the mutualist's buffering capacity is exceeded.

From our central findings, we draw two main conclusions. First, positive interactions can indeed enhance ecosystem resistance to and recovery from large-scale disturbance as has been proposed, but not yet demonstrated in a real, non-theoretical system[13,30]. As mutualism-dependent ecosystems occur in all corners of the earth (for example, mangroves, seagrass meadows, coral reefs, peat bogs, boreal forests), we suspect that similar processes may enhance the stress resistance and accelerate the recovery of critical habitat-forming species globally. Second, we anticipate that managers may achieve impressive gains in ecosystem resilience through relatively little investment where they integrate keystone mutualisms and optimal patch distributions into conservation and restoration strategies[26]. To our knowledge, these analyses are the first to show the potential for spatially dispersed, stress-resistant patches to promote ecosystem resilience to a major disturbance, a finding that may help explain differences in resilience observed across ecosystems and enhance our ability to maintain ecosystems in the face of climate change.

## Methods

**Latitudinal survey.** To evaluate our hypothesis that the presence of mussel mutualists increases cordgrass resistance to drought, we assessed cordgrass survival within die-off areas that formed during the 2010–2012 drought near its conclusion in June 2012 (Supplementary Fig. 1). We selected sites based on their distribution across the region that NOAA drought records indicated was significantly impacted by this drought event: a ~250 km stretch of coastline spanning from southern Georgia to central South Carolina (Palmer Drought Severity Index values were $< -2$ for 15 of 18 months in South Carolina ($-3.5 \pm 1.3$, mean PDSI value $\pm$ STD) and 18 of 18 months in Georgia ($-4.1 \pm 0.6$), NOAA National Center for Environmental Information: www.ncdc.noaa.gov/temp-and-precip/ drought/historical-palmers/). We used Google Earth to identify 13 sites that contained relatively large ($>1,000\,m^2$) marsh platforms, which occur at higher elevation, interior marsh regions where drought is most likely to dry and oxidize otherwise moist, anoxic marsh soils[5]. At nine of these sites, we observed three to eleven die-off areas with standing dead shoots and dry, cracked and/or salt-crusted soil, suggesting that cordgrass had senesced recently and that soil drying was a contributing factor to the observed die-off[17]. For each die-off encountered, we used a transect tape to measure its average length and width. From these dimensions, we estimated the die-off area ($A_{\text{Die-off}}$) using the equation for an oval. We excluded die-offs that abutted docks, causeways or woody vegetation fringing the terrestrial marsh border, or those associated with wrack (that is, dead plant material mats covered $>3\%$ of the die-off area), as they were unlikely to have been generated by drought[31]. Using March 2010 Google Earth Imagery, we then verified that each of our 53 die-off areas had been vegetated before 2010–2012 and, thus, formed during this severe drought period.

In each die-off, we measured the area of every surviving cordgrass patch, noted whether it was associated with mussels ($C_{Mussels}$) or not ($C_{No\ mussels}$), and determined the area of dead cordgrass associated with mussels ($D_{Mussels}$, Supplementary Fig. 2). A patch was scored as being 'associated with mussels' if 1 or more adult ($\geq 60\ mm$ shell length) mussels were observed embedded in the mud $<10\ cm$ from the base of live stems that constituted the patch. On the basis of experimental results showing that effects of mussels on cordgrass growth are localized (that is, up to $\sim 10\ cm$ distance, see the 'Spatial extent of the mussel-cordgrass interaction' section below), we assume the effect of mussels on cordgrass during drought is restricted to this spatial scale and consistent across study sites. Next, we summed $A_{Die-off}$, $C_{Mussels}$, $C_{No\ Mussels}$ and $D_{Mussels}$ measures across all die-offs surveyed at each site and calculated the probability of cordgrass surviving when associated with mussels as: $P_{Mussels} = (\Sigma C_{Mussels})/(\Sigma C_{Mussels} + \Sigma D_{Mussels})$, and when not associated with mussels as: $P_{No\ Mussels} = (\Sigma C_{No\ Mussels})/(\Sigma C_{No\ Mussels} + \Sigma A_{Die-off} - \Sigma C_{Mussels} - \Sigma D_{Mussels})$. We pooled cordgrass survival data from all die-offs surveyed at each site (rather than analysed each die-off separately) because these die-off areas were often positioned in close proximity to one another ($<30\ m$ between adjacent die-offs). Therefore, they were unlikely to represent independent cordgrass mortality events. Finally, to test our hypothesis that cordgrass survival was indeed higher when associated with mussels, we analysed the effect size and significance of Mussel Presence on the probability of survival using a linear model in R[32]. We visually inspected the distribution of residuals to verify that the model assumption of homogeneity in variance was met.

Finally, we returned to 7 of the 9 die-off sites in October 2014 to survey all mounds observed within 5, $1 \times 20\ m$-long transects positioned outside of die-off areas (in regions of the marsh platform where cordgrass survived the drought) to assess whether the spatial cover of mussels was similar within die-off areas compared with adjacent marsh habitat. Mussel mounds covered between 0.21 and 1.1% of transect area across the 7 sites, indicating coverage of the mutualism was indeed similarly low in marsh platform areas experiencing cordgrass die-off compared with those areas that survived this severe drought.

**Spatial extent of the mussel–cordgrass interaction.** To assess the spatial extent of mussel effects on soil structure and plant growth, we established six $0.25\ m^2$ plots in a Sapelo Island, GA salt marsh platform in April 2013 and added 20 mussels to half (the same plots where we tested mussel effects on salinity and aboveground biomass, see the main text and the 'Mussel effects on porewater salinity and cordgrass growth' section below). Immediately after transplanting the mussels, we measured the diameter of each mussel mound in both the North–South and East–West directions. Then, in mid-August, we measured the diameter of the elevated pseudofeces layer that had accumulated around the mussel mound in each direction. To assess the lateral extent of each mound's influence on soil structure, we subtracted the initial mussel mound diameter from the pseudofeces layer diameter in each direction, averaged these values, and divided this number by two (N–S and E–W diameters) for each plot. We found that the pseudofeces layer extended an average of $12.9 \pm 1.5\ cm$ (mean $\pm$ s.e.m.) from the edge of the mussel mound. These results indicate that the 10-cm cut-off used to assign patches as associated or not associated with mussels in the latitudinal survey provided a conservative estimate of the lateral extent of the cordgrass–mussel interaction.

We also measured the height of eight cordgrass stems emerging within the pseudofeces layer and the marsh platform immediately adjacent to the pseudofeces layer to determine if cordgrass growth is enhanced within the pseudofeces footprint. We then used a two-sided $t$-test to assess if average stem height differed on and off the pseudofeces layer. Cordgrass stems were 12 cm taller, on average, on than off the layer, although this effect was not statistically significant ($P = 0.1$, Supplementary Fig. 3).

**Observational evidence of mussel effects on soil water storage and salinity stress.** To examine whether mussel aggregations enhance soil water storage, we extracted $8 \times 20\ cm$ (diameter $\times$ depth) soil cores from naturally occurring mussel aggregations and from no mussel, control plots from the Oakdale Creek marsh platform on Sapelo Island, GA ($N = 20$ replicates per plot type). Cores were collected during a neap tide period when the marsh platform had not been inundated by the tides or been rained on for 3 consecutive days. In the lab, aboveground vegetation was removed from each core, after which the core was weighed (wet weight), oven-dried at $60\ °C$ for 96 h and reweighed (dry weight). From these measures, we calculated soil water content as: $100 \times$ (wet weight – dry weight)/(dry weight). After verifying that data met assumptions of normality using a Shapiro–Wilk's test, we analysed the effect of mussels on soil water content with a two-sided $t$-test.

To investigate if a possible mechanism by which mussels increase cordgrass drought resistance is that they prevent soils from drying, we monitored porewater salinity in cordgrass' rooting zone. In May 2012, we set 11 pairs of lysimeters[20] to a depth of 25 cm to span cordgrass' root zone: one lysimeter was positioned in the middle of a mussel mound (7–22 mussels per mound) and another 1 m away in a control (no mussel) location. Lysimeters were distributed across 6 hectares of Sapelo Island cordgrass-dominated marsh platform. Once per week from 15 May through 14 August 2012, we recorded the salinity of porewater extracted from each lysimeter. To summarize the relationship between mussels and salinity over this monitoring period, we calculated the difference in salinity between each

lysimeter pair for each date. We then averaged these values to assess the mean difference in salinity across all pairs, and used a one-sample $t$-test to analyse whether the mean difference in salinity across all lysimeter pairs over this time period was significantly different from zero. To evaluate if mussel aggregations buffer changes in porewater salinity more as marsh soils become drier, we calculated for each date: (1) the average salinity in the 10 control wells as a measure of the ambient marsh salinity; and (2) the average difference in salinity between each mussel aggregation and control well pair as a measure of the buffering capacity of the mussels. Using linear regression, we investigated the relationship between these two variables.

**Mussel effects on porewater salinity and cordgrass growth.** To test if mussels reduce soil salinity stress and enhance aboveground cordgrass biomass, we established six $0.25\ m^2$ plots in a Sapelo Island, GA salt marsh platform in April 2012. We transplanted 20 mussels in an aggregated manner to mimic natural mounds in three randomly chosen plots and agitated the soil surface to account for disturbance effects of mussel transplantation in the other three 0-mussel control plots. We added, rather than removed mussels, from cordgrass patches because extracting mussels intensively disturbs plant roots and soils, and because mussels' residual impacts on plant and soil conditions are long lasting. During a neap tide in mid-August 2013 when the marsh platform was not inundated by the tides and did not receive any freshwater input from precipitation for 3 days, we inserted rhizon porewater samplers in the surface (0–5 cm) and sub-surface (5–15 cm) cordgrass rooting zones in each plot. The salinities of the porewater samples were then measured using a handheld refractometer. In October 2013, we collected, cleaned, dried and weighed all cordgrass stems from each plot to assess aboveground biomass. After verifying that data met assumptions of normality using Shapiro–Wilk's tests, we analysed differences between mussel addition and control plots using two-sided $t$-tests.

**Cordgrass recolonization dependence on clonal growth.** To assess whether seed dispersal, clonal expansion or both are reproductive modes by which cordgrass recolonizes die-off mudflats, we marked 20, $1\ m^2$ plots immediately adjacent to cordgrass monocultures bordering a die-off on Sapelo Island, GA in May 2008. An drought lasting from 2006 to late 2007 generated the $1,200\ m^2$ mudflat within which this experiment took place. We assigned each plot one of two treatments: clonal ramet exclusion or control ($N = 10$ replicates per treatment). To prevent cordgrass from colonizing via vegetative growth and thus isolate the contribution of seeds to recovery, we used a flat shovel to install landscape fabric around the perimeter of each clonal ramet exclusion plot to a depth of 30 cm, a depth below which cordgrass rarely produces rhizomes at higher marsh platform elevations. Control plots were trenched with the shovel to account for disturbance effects. In July 2009, we determined per cent cordgrass cover using a 100-cell frame positioned over each plot. In July 2010 and October 2011, we modified our method to better assay the source of emergent shoots: in each plot, we tugged on each shoot present to test whether it was anchored by a deep-penetrating rhizome, indicating it was a clonal ramet, or not, indicating a seedling, and recorded the number of each. We manually removed any ramets that emerged in clonal ramet exclusion plots ($<10$ total shoots across all exclusion plots and years). We analysed the effect size and significance of Ramet Treatment using a two-sided $t$-test on the per cent cover data from 2009. As we did not observe a single cordgrass seedling in any plot over the duration of the experiment, multivariate analyses that incorporate responses of both cordgrass seeds and ramets were unnecessary. Thus, we analysed the effect size and significance of Ramet Treatment over time on the number of ramets per plot in 2010 and 2011 using repeated-measures analysis of variance.

**Mussel effects on recovery at the patch scale.** To evaluate whether the lateral expansion of cordgrass patches increases with the number of mussels within a patch and if the effect of mussel density is more important than other factors such as elevation and snail (Littoraria irrorata) grazer density[20] on local (patch) scales, we counted the number of mussels within each of 79 naturally occurring remnant patches and measured the radius of each patch in both East–West and North–South directions in late May 2012. We measured the elevation and position of each plot using an RTK GPS (Trimble R6 GNSS, $\pm 1.5\ cm$ vertical and $\pm 1\ cm$ horizontal accuracy) and placed a $0.25\ m^2$ sampling frame over the centre of each plot within which we counted the number of snails. Remnant patches were distributed in recently formed die-off areas located at two Sapelo Island, GA marsh platform sites. In November 2012, June 2013 and December 2013 (after 6, 12 and 19 months, respectively), we re-measured the patch radii and counted snails. Patches that had merged with other cordgrass patches or cordgrass bordering die-off areas were not monitored after 12 and 19 months, resulting in only 71 and 64 patches being monitored at these dates. We then visually examined the data to ensure assumptions of normality were met and used a linear mixed effect model to investigate the effect size and significance of elevation, mussel density, average snail density and their interactions on patch aerial expansion (that is, the change in patch area) at each monitoring interval.

**Modelling marsh recovery at the landscape scale.** To assess the relative importance of recolonization from cordgrass bordering the die-offs and cordgrass residing in surviving patches in regulating marsh recovery, we developed a cellular automaton-based model[33]. To develop cordgrass expansion metrics for the model, we monitored 0.25-m$^2$ plots (an individual cell in the model) in healthy cordgrass stands ($N = 10$), mudflat areas adjacent to patches ($N = 79$), mudflat areas adjacent to cordgrass stands that bordered die-offs ($N = 69$) and mudflat areas >2 m from a cordgrass patch or border source ($N = 10$) over 12 months. All plots were located in Sapelo Island, GA marsh platforms. Mudflat plots were deemed transitioned to cordgrass if the cordgrass stem density was >50% of the average stem density observed in ten replicate 0.25 m$^2$ plots positioned in healthy cordgrass stands located adjacent to die-off areas. The probability of a mudflat cell transitioning to cordgrass if adjacent to a neighbouring patch, border or mudflat cell was calculated as the proportion of monitoring plots that transitioned from each source. On the basis of our empirical results, we adopted the following transition probabilities in the model. Mudflat cells had a colonization probability of 0.7-times the number of the neighbouring cordgrass patch cells, 0.5 times the number of cordgrass border cells and 0 times the number of neighbouring mudflat cells (that is, none of the monitored mudflat plots located >2 m from a cordgrass patch or border source transitioned to cordgrass). Next, we used a GPS to collect die-off size and patch location data from nine Sapelo Island die-offs and used Corrected Ripley's K analyses[25] to characterize the dispersion of patches within each of the six surveyed mudflats with >10 patches. Patches exhibited random and clustered distributions in these die-offs (see Supplementary Fig. 6 for examples of naturally occurring patch distribution patterns).

We used the model to simulate marsh recovery under 25 realistic scenarios that included all combinations of five mudflat sizes (50, 500, 1,000, 1,500 and 2,000 m$^2$ which correspond to square grids with 28, 89, 126, 155, and 179, 0.25 m$^2$ cells per side) and five patch distributions (see the main text). From three replicate simulations of each scenario, we calculated the average marsh recovery interval (that is, the number of time steps (years) until 95% of initial mudflat cells transitioned to cordgrass) and the per cent of die-off recovered from patches.

**Effect of patch density on recovery at the die-off scale.** To evaluate if die-offs close faster with increasing patch density and assess the model's validity, we counted the mussel-associated patches within and measured using an RTK GPS the area of bare mudflat in seven Sapelo Island die-offs in both June 2012, at the drought's conclusion, and July 2015, after 3 years of recovery. We assessed the relationship between mussel-associated patch density and the proportional change in mudflat area from 2012 to 2015 using linear regression (Supplementary Fig. 8). We visually inspected the distribution of residuals to verify that the model assumption of homogeneity in variance was met.

**Data availability.** Survey and field experiment data supporting these findings are freely available online (Angelini *et al.* A keystone mutualism underpins resilience of a coastal ecosystem to drought. Dryad Digital Repository, http://dx.doi.org/10.5061/dryad.d875g (2016)) and from the corresponding author on request. R code for the marsh recovery model is also available on request.

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

## Acknowledgements

We thank SINERR and GCE LTER for providing logistical support. Funding was provided by NSF GRFP (DGE-0802270) and NSF DEB (No. 1546638) to C.A.; NSF Career (No. 1056980) to B.R.S.; EU Marie Curie Career Integration Grant (FP7 MC CIG 61893) to J.N.G. and NWO-VENI (No.863.12.003) to T.v.d.H.

## Author contributions

C.A., J.N.G., T.v.d.H., L.P.M.L., A.J.P.S., J.v.d.K. and B.R.S. conceived and designed the experiments; C.A., J.N.G., J.v.d.K., M.D.-H., L.P.M.L., T.v.d.H. and B.R.S. performed the experiments; C.A. and J.v.d.K. analysed the data; B.R.S. and J.v.d.K. contributed materials/analysis tools; C.A., J.N.G., L.P.M.L., J.v.d.K., T.v.d.H. and B.R.S. co-wrote the paper.

## Additional information

**Competing financial interests:** The authors declare no competing financial interests.

