## [Peer review file · Nature Communications]

Reviewers' Comments:

Reviewer #1 (Remarks to the Author)

The authors do extensive, creative, and novel field work, experimental manipulations and modeling, to show that "spatially dispersed, stress-resistant patches to promote ecosystem resilience to a major disturbance". Their stress-resistant patches are ramets of cordgrass in salt marsh colonized by mussels, which ameliorate salinity and toxic stresses for plants by helping to retain water locally. The authors have surveyed die-offs in marshes along 250 km of shoreline along the Eastern US coast. They manipulated presence and absence of mussels within tussock patches, and observed soil conditions and grass performance. They noted only vegetative spread, rather than recolonization by seed, a pattern often noted in marsh vegetation. They explored die-off size and patch dispersion effects with a cellular automata simulation. Overall, they have made a very valuable contribution to our knowledge of both marsh conservation and restoration, and more fundamental understanding of the role of spatial patterns and mutualisms in ecosystem resilience.

I've made some specific comments in red on the attached pdf.

Reviewer #2 (Remarks to the Author)

This is an interesting paper linking positive interactions, or facilitation, among organisms to large scale ecosystem or landscape resilience. This is excellent. However, it is difficult for me to assess exactly how much evidence is actually marshalled for real facilitative interactions between mussels and cordgrass, and drought as a mechanism. If I understand correctly, 1) there is a strong spatial correlation between mussels and cordgrass over much of the southeastern coastline and in particular after dieback. Please note that this could be caused by mussels cordgrass both occupying favorable microsites, and this would not be facilitation. 2) Experimental planting and addition of mussels (this is a great experiment) showed lower soil salinity with mussels present. 3) There is no evidence that mussels or measured changes in salinity increase the survival, growth, or spread of cordgrass. 4) A cellular automaton model, not mentioned or explained in the Supplementary Methods other than a Table, found that cordgrass expansion was facilitated by mussels. If there is more evidence than this, then it should be presented in a more linear way, and I apologize for missing it. If this is all of the evidence for your fundamental thesis, then the results are not strong. Your correlative patterns and evidence for mussels improving salinity conditions is great, but without basis experimental field evidence that mussels improve the fitness, growth, clonal expansion, or some aspect of the performance of cordgrass the overall results are not compelling.

There are no line numbers so I apologize for any vagueness in the following comments.

The sentence at the end of page 3 and beginning of page 4 is a little awkward, perhaps break into two.

What is a marsh platform?

When you describe cordgrass' rooting zone, can you add the depth of this zone here in the text? Also the structure of the sentence is odd.

May I suggest something like: "After 5 months, we found that porewater salinity, an integrative measure of soil moisture content (11), in the rooting zone of cordgrass (0-25 cm below the soil surface), fell from 45 parts per thousand (ppt) in control, no mussel treatments to 37ppt in mussel addition treatments (T-test: $P < 0.0001$, Fig. S4)." Or break this into two sentences for even more clarity.

You note that "These results collectively indicate that mussels increase cordgrass survival..." but I can find no results for the relationship between mussels and cordgrass survival. See overall comments at the beginning of this review.

This same paragraph contains information from both experimental and correlative patch measurements. This is fine and the results are interesting, but why weren't all of these measurements done in the experimental plantings, or in both experimental and "natural" conditions? Also, in this same paragraph, is the difference between 45 and 37 ppt biologically significant? For example, ocean water is about 35 ppt, will an increase of 8 ppt dramatically affect cordgrass? I would think so, Vasquez et al. (2006; American Journal of Botany) found that cordgrass could grow in salinities about that of seawater. But at that salinity growth was reduced quite a bit. I'd guess further decreases would be bad. This is not a crucial thing, but it might be worthwhile to address this pretty important aspect of cordgrass biology in the context of your work. Also, the figures referred to here and throughout text did not come out right in the PDF.

In the paragraph describing patch expansion it would be good to include the length of time these patches were monitored.

Reviewer #3 (Remarks to the Author)

Review of the ms A keystone mutualism underpins salt marsh resilience to drought

The ms is well written, the topic examined is timely and the dataset is appropriate for exploring the questions addressed. I think I know the question explored rather well but as I am coming from a numerical background I will focus my comments on the methods regarding statistical analyses and simulations. Overall, I very much would like to see the paper published. However I do think that some moderate revisions are needed in order to show the result neatly and in a rigorous scientific way. The amendments that I suggest require only some additional work regarding the statistics and model description but no extra field work, lab work, or indeed no additional data are required. To that end the revisions should be feasible in moderate to fairly short time frames.

1. My first concern regards the way the probability of survival in the presence and in the absence of mussels is defined: vegetation patches were classified as associated with mussels if one or more stem was positioned < 10 cm from a mussel. This is a rather discrete way of defining plant-mussel associations as the zone of influence of 10 cm is rather arbitrary firstly because plants at e.g. 12 cm could also potentially benefit from mussels and then secondly because the spatial distribution of mussel patches influences survival as the authors acknowledge in the ms and perform point pattern analysis (i.e. aggregation seems to be more often than not beneficial for survival). Overall a formula that incorporates space in a more continuous manner in probability of survival would be beneficial.

2. While the addition of the cellular automaton model is certainly making the ms stronger at present the model is not properly described. First the authors need to list the actual simulation space employed that includes (a) grid size, (b) initialization across all scenarios, and (c) simulation time steps. Currently they only provide info regarding different patch sizes, and the time steps that it took to recover. The reader needs to know both the grid size, and how the values in cells were initially distributed. Then (d) it is very important for the reader to know the actual transition probabilities - the authors mention that they were calculated, but no values are listed and thus the model is not reproducible from an external reader. My final question here is why did you actually multiply with 0.75? The fact that this was the proportion of surviving patches associated with mussels from survey data should already have been incorporated in your transition probabilities of the cellular automaton model. Unless I am missing something, these should be an independent result of the model, while currently it seems rather like superimposing the result in the model.

3. Point pattern analysis: The authors do well to use Ripley's K function to examine the spatial distribution of remnant patches. However my view is that these results are not properly described

and discussed in the ms: From Fig. S7 it is evident that 3 out of 4 sites have an aggregated distribution across small to intermediate scales, while the remaining 1 has a distribution that may not be separated from complete spatial randomness at fine scales and it is aggregated at medium to coarser scales. Thus remnant patches are aggregated at the scales where the actual interaction is taking place. While this result is in part also in the cellular automata model results, my view is that this spatial analysis needs to be better incorporated and discussed in the main text. Currently the authors mention that 'While patches exhibited a nearly uniform, or even, distribution in one die-off, they exhibited random and clustered distributions in others'. Unless this text derives from plots or data not presented here, the text is not corresponding to your figure S7. I think that the aggregation pattern needs to be more emphasized and discussed in the text.

4. Minor comment: please remove all '!' symbols from your figure legends in the supplement

Reviewers' comments:

Reviewer #1 (Remarks to the Author):

The authors do extensive, creative, and novel field work, experimental manipulations and modeling, to show that "spatially dispersed, stress-resistant patches to promote ecosystem resilience to a major disturbance". Their stress-resistant patches are ramets of cordgrass in salt marsh colonized by mussels, which ameliorate salinity and toxic stresses for plants by helping to retain water locally. The authors have surveyed die-offs in marshes along 250 km of shoreline along the Eastern US coast. They manipulated presence and absence of mussels within tussock patches, and observed soil conditions and grass performance. They noted only vegetative spread, rather than recolonization by seed, a pattern often noted in marsh vegetation. They explored die-off size and patch dispersion effects with a cellular automata simulation. Overall, they have made a very valuable contribution to our knowledge of both marsh conservation and restoration, and more fundamental understanding of the role of spatial patterns and mutualisms in ecosystem resilience.

I've made some specific comments in red on the attached pdf.

We appreciate these complementary and encouraging comments and are happy to learn that he/she found much value in our research. We have responded to the comments that this reviewer made in the pdf that was attached to the editor's letter. Of note, we corrected the highlighted verb tense issues, now present the rationale for the time steps used in our cellular automaton model, and clarified the language we use in interpreting the model results (see additional comments below in response to Reviewers 2 and 3). In particular, we expand our description of the model and its assumptions in Lines 156-188 to ensure our readers can interpret our findings in light of these details without having to refer to the Supplemental Information.

Reviewer #2 (Remarks to the Author):

This is an interesting paper linking positive interactions, or facilitation, among organisms to large scale ecosystem or landscape resilience. This is excellent. However, it is difficult for me to assess exactly how much evidence is actually marshalled for real facilitative interactions between mussels and cordgrass, and drought as a mechanism. If I understand correctly, 1) there is a strong spatial correlation between mussels and cordgrass over much of the southeastern coastline and in particular after dieback. Please note that this could be caused by mussels cordgrass both occupying favorable microsites, and this would not be facilitation. 2) Experimental planting and addition of mussels (this is a great experiment) showed lower soil salinity with mussels present. 3) There is no evidence that mussels or measured changes in salinity increase the survival, growth, or spread of cordgrass. 4) A cellular automaton model, not mentioned or explained in the Supplementary Methods other than a Table, found that cordgrass expansion was facilitated by mussels. If there is more evidence than this, then it should be presented in a more linear way, and I apologize for missing it. If this is all of the evidence for your fundamental thesis, then the results are not strong. Your correlative patterns and evidence for mussels improving salinity conditions is great, but without basis experimental field evidence that mussels improve the fitness, growth, clonal

expansion, or some aspect of the performance of cordgrass the overall results are not compelling.

We appreciate that the reviewer brought these important concerns to our attention and, upon re-reading our initial submission, we completely agree. To address these concerns, we have: 1) re-arranged the presentation of our results to improve the flow of logic, 2) inserted additional experimental evidence to help validate our assertions, 3) more clearly related our findings to prior research investigating the effects of elevated salinity on cordgrass fitness to reinforce the conclusions we draw from our results; and 4) provided a more comprehensive description of the cellular automaton model in the Main Text and Methods to help the reader visualize and interpret the simulations we ran.

In particular, we now first present our observational evidence that mussels enhance water storage and buffer against increases in salinity (these results were presented at a later point in the initial submission), and then summarize results from a field experiment showing that mussels reduce porewater salinity and stimulate cordgrass biomass. These unpublished data from a recent study of ours are detailed in Lines 124-136, Lines 290-306, and in Fig. 2C and 2D. In adding these experimental data, we address the key issue this reviewer highlights that we provided no “basis experimental field evidence that mussels improve the fitness, growth, clonal expansion or some aspect of the performance of cordgrass” in the initial submission, and demonstrate that the positive relationship between mussels and surviving cordgrass observed in the latitudinal survey is not simply a result of these organisms occupying favorable microsites.

Importantly, we then compare our observational and experimental results to those reported in Silliman et al. 2005, Science, reporting in Lines 129-131 that “In both the experiment and weekly well monitoring, salinity only reached levels shown to be associated with cordgrass mortality during drought (>48ppt)⁴ in marsh areas without mussels”. In essence, our results indicate that mussels prevented salinities from reaching levels that an experiment and observations in the Silliman et al. study showed dramatically reduce cordgrass growth and survival fitness metrics in this study system. Finally, we synthesize these findings by stating in Lines 136-140, “Together with the survey results indicating that mussel mounds substantially enhance cordgrass survival within drought-induced die-offs, these experimental and observational findings provide evidence that mussel amelioration of soil stress, potentially in combination with their enhancement of cordgrass growth before drought occurs, is increasing cordgrass resistance to drought across this region.”

Thus, we feel that we do a much better job of integrating our field observations, experimental results, and the peer-reviewed literature in the revised manuscript to build a straight-forward, logical case that mussels and cordgrass are engaged in a facultative mutualism and that this positive interaction has widespread consequences for cordgrass survival (and hence fitness) during drought.

Finally, we realized that several key details regarding the cellular automaton model were omitted in the main text of the initial submission – especially those regarding the grid size, time step, initialization scenarios, cell transition probabilities, and assumptions – and now include these in the Main Text and Methods. Further details are provided in response to Reviewer 3 below.

There are no line numbers so I apologize for any vagueness in the following comments. *We apologize, and have added line numbers to the revised manuscript.*

The sentence at the end of page 3 and beginning of page 4 is a little awkward, perhaps break into two.

We have broken this long sentence into two, more easily digestible sentences. Please see Lines 62-65.

What is a marsh platform?

We now add the following description, "the higher elevation, interior marsh platforms" to help orient the reader to the areas in the marsh where the die-offs occur in Line 72.

When you describe cordgrass' rooting zone, can you add the depth of this zone here in the text? Also the structure of the sentence is odd.

We now define the surface (0-5cm) and sub-surface (5-15cm) levels of the cordgrass rooting zone in Lines 125-127 and reworded this sentence.

May I suggest something like: "After 5 months, we found that porewater salinity, an integrative measure of soil moisture content (11), in the rooting zone of cordgrass (0-25 cm below the soil surface), fell from 45 parts per thousand (ppt) in control, no mussel treatments to 37ppt in mussel addition treatments (T-test: $P < 0.0001$, Fig. S4)." Or break this into two sentences for even more clarity.

In reorganizing the results, we removed these sentences.

You note that "These results collectively indicate that mussels increase cordgrass survival..." but I can find no results for the relationship between mussels and cordgrass survival. See overall comments at the beginning of this review.

We have completely re-written this paragraph and moved it later in the presentation of our results to clarify our conclusions. Please see our detailed comments to the reviewer's general comments above and Lines 124-140 in the text.

This same paragraph contains information from both experimental and correlative patch measurements. This is fine and the results are interesting, but why weren't all of these measurements done in the experimental plantings, or in both experimental and "natural" conditions?

We have restructured this section of the manuscript and now dedicate one paragraph to our observational results and one to our experimental results to clarify the measurements collected in each study.

Also, in this same paragraph, is the difference between 45 and 37 ppt biologically significant? For example, ocean water is about 35 ppt, will an increase of 8 ppt dramatically affect cordgrass? I would think so, Vasquez et al. (2006; American Journal of Botany) found that cordgrass could grow in salinities about that of seawater. But at that salinity growth was reduced quite a bit. I'd guess further decreases would be bad. This is not a crucial thing, but it might be worthwhile to address this pretty important aspect of cordgrass biology in the context of your work.

We now report on measured differences in porewater salinities that have been found to drive significant variation in cordgrass production and survival in prior experimental and observational studies (see Silliman et al. 2005, Science) in Lines 129-131. Importantly, we note that we only recorded salinities in our weekly well monitoring and field experiment that have been shown in the Silliman et al. study to be associated with significantly reduced cordgrass biomass and enhanced mortality in marsh areas without

mussels. See additional details in our response to this reviewer's general comments above.

Also, the figures referred to here and throughout text did not come out right in the PDF. *We apologize that the figures did not come through correctly and have uploaded them in an alternative format in the revised submission.*

In the paragraph describing patch expansion it would be good to include the length of time these patches were monitored.

We now clearly state that the experimental and natural patches were monitored for one year.

Reviewer #3 (Remarks to the Author):

Review of the ms A keystone mutualism underpins salt marsh resilience to drought

The ms is well written, the topic examined is timely and the dataset is appropriate for exploring the questions addressed. I think I know the question explored rather well but as I am coming from a numerical background I will focus my comments on the methods regarding statistical analyses and simulations. Overall, I very much would like to see the paper published. However I do think that some moderate revisions are needed in order to show the result neatly and in a rigorous scientific way. The amendments that I suggest require only some additional work regarding the statistics and model description but no extra field work, lab work, or indeed no additional data are required. To that end the revisions should be feasible in moderate to fairly short time frames.

We thank the reviewer for these supportive remarks and his/her particular focus on the statistical analyses and simulations, as these comments complement those made by the first two reviewers and challenged us to improve the clarity of these important components of our study.

1. My first concern regards the way the probability of survival in the presence and in the absence of mussels is defined: vegetation patches were classified as associated with mussels if one or more stem was positioned < 10 cm from a mussel. This is a rather discrete way of defining plant-mussel associations as the zone of influence of 10 cm is rather arbitrary firstly because plants at e.g. 12 cm could also potentially benefit from mussels and then secondly because the spatial distribution of mussel patches influences survival as the authors acknowledge in the ms and perform point pattern analysis (i.e. aggregation seems to be more often than not beneficial for survival). Overall a formula that incorporates space in a more continuous manner in probability of survival would be beneficial.

This is an important concern and we agree that we did not adequately justify our rationale for assigning patches as being 'associated' or 'not-associated' with mussels based on this discrete, 10-cm distance in the initial submission. To correct this, we now provide results from an experiment in which we added mussel mounds to a marsh platform and monitored the lateral extent of their effects on soil structure and cordgrass growth in the supplement and refer to these results in Lines 105-106 of the Main Text.

With regard to the second part of this remark regarding the pattern of patch survival, we are unclear on exactly what his/her concern is, but have revised several lines of text to clarify and expand upon our rationale for exploring recovery under the different patch

distribution scenarios (Lines 179-186). In particular, we now articulate that we investigated uniform patch distributions to compare the recovery trajectories of natural die-off areas (i.e. those with no patches or patches in random or clustered distributions) to die-off areas planted with uniform arrays of grass transplants, the typical approach adopted by many restoration practitioners.

Finally, we opted not to use a continuous model for salt marsh expansion as the cellular automaton approach is very simple and transparent, and allows for a direct link to our monitoring data. As the main conclusions are not dependent on the absolute rate but rather on the relative differences in the rate of recovery and our monitoring of natural patch expansion revealed that mussels only subtly influence patch expansion and that this effect fades after 19 months (Fig. S5, Table S2), we felt that enhancing the complexity of the model to incorporate a continuously decreasing influence of mussels around a patch (while mathematically feasible using a convolution technique) would not significantly alter the conclusions or enhance the rigor of our study.

2. While the addition of the cellular automaton model is certainly making the ms stronger at present the model is not properly described. First the authors need to list the actual simulation space employed that includes (a) grid size, (b) initialization across all scenarios, and (c) simulation time steps. Currently they only provide info regarding different patch sizes, and the time steps that it took to recover. The reader needs to know both the grid size, and how the values in cells were initially distributed. Then (d) it is very important for the reader to know the actual transition probabilities - the authors mention that they were calculated, but no values are listed and thus the model is not reproducible from an external reader. My final question here is why did you actually multiply with 0.75? The fact that this was the proportion of surviving patches associated with mussels from survey data should already have been incorporated in your transition probabilities of the cellular automaton model. Unless I am missing something, these should be an independent result of the model, while currently it seems rather like superimposing the result in the model.

We have edited and expanded this section of the manuscript as well as the methods that support it to clarify the details of our cellular automaton model. We now present explicit details on the grid size, initialization for each scenario, transition probabilities, model assumptions, and the time steps, which are all based on field-collected data. These edits can be found in Lines 159-188 in the Main Text and Lines 326-348 in the Methods.

In addition, we no longer calculate the contribution of mussel-associated patches to recovery by using this 0.75 multiplier. Instead, we simply report the contribution of patches in general to recovery, a far more straightforward and accurate approach for describing the model results.

3. Point pattern analysis: The authors do well to use Ripley's K function to examine the spatial distribution of remnant patches. However my view is that these results are not properly described and discussed in the ms: From Fig. S7 it is evident that 3 out of 4 sites have an aggregated distribution across small to intermediate scales, while the remaining 1 has a distribution that may not be separated from complete spatial randomness at fine scales and it is aggregated at medium to coarser scales. Thus remnant patches are aggregated at the scales where the actual interaction is taking place. While this result is in part also in the cellular automata model results, my view is that this spatial analysis needs to be better incorporated and discussed in the main text. Currently the authors mention that "While patches exhibited a nearly uniform, or even,

distribution in one die-off, they exhibited random and clustered distributions in others'. Unless this text derives from plots or data not presented here, the text is not corresponding to your figure S7. I think that the aggregation pattern needs to be more emphasized and discussed in the text.

We agree that the spatial pattern analyses were not adequately or accurately presented in the main text of the initial submission and are critical for informing the design of the model simulations as the reviewer highlights. To address this concern, we expanded our discussion in the main text and added the following sentences in Lines 179-186: "We explored random and clustered patch patterns because Ripley's K analyses²⁵ of field-collected data indicate patches surviving within die-off areas exhibit these spatial configurations (Fig. S6, see Methods for details). Likewise, we simulated the recovery of die-offs with uniformly distributed patches to compare cordgrass recolonization of naturally-occurring die-offs (i.e. those with only cordgrass bordering, or with cordgrass bordering and in random or clustered patches) to restoration projects that typically plant uniform – i.e. regularly spaced – arrays of marsh grass plugs within denuded areas²⁶."

4. Minor comment: please remove all '!' symbols from your figure legends in the supplement

We have revised the figure legends in the supplement.

Reviewers' Comments:

Reviewer #2 (Remarks to the Author)

Reviewer #3 (Remarks to the Author)

I have gone through the revised ms version and according to my opinion all of my comments have been addressed.

In particular the selection of 10 cm as a cut off has been justified and the description, initialization and simulations of the cellular automaton model makes the model both sound and reproducible.

I am satisfied with the ms on its current form